# A New Stress-Based Formulation for Modeling Notched Fiber-Reinforced Laminates

**DOI:** 10.3390/polym14245552

**Published:** 2022-12-19

**Authors:** Xian Liu, Linxin Wang, Quantian Luo, Zhonghao Bai, Qing Li, Jian Hu

**Affiliations:** 1State Key Laboratory of Advanced Design and Manufacture for Vehicle Body, Hunan University, Changsha 410082, China; 2Hunan Provincial Key Laboratory of Fine Ceramics and Powder Materials, School of Materials and Environmental Engineering, Hunan University of Humanities, Science and Technology, Loudi 417000, China; 3School of Mechanical and Mechatronic Engineering, University of Technology Sydney, Ultimo, NSW 2007, Australia

**Keywords:** carbon-fiber-reinforced plastic (CFRP) laminates, notched plate, digital image correlation (DIC), finite element modeling, damage mechanism

## Abstract

Laminated plates are often modeled with infinite dimensions in terms of the so-called Whitney–Nuismer (WN) stress criteria, which form a theoretical basis for predicting the residual properties of open-hole structures. Based upon the WN stress criteria, this study derived a new formulation involving finite width; the effects of notch shape and size on the applicability of new formulae and the tensile properties of carbon-fiber-reinforced plastic (CFRP) laminates were investigated via experimental and theoretical analyses. The specimens were prepared by using laminates reinforced by plain woven carbon fiber fabrics and machined with or without an open circular hole or a straight notch. Standard tensile tests were performed and measured using the digital image correlation (DIC) technique, aiming to characterize the full-field surface strain. Continuum damage mechanics (CDMs)-based finite element models were developed to predict the stress concentration factors and failure processes of notched specimens. The characteristic distances in the stress criterion models were calibrated using the experimental results of un-notched and notched specimens, such that the failure of carbon fiber laminates with or without straight notches could be analytically predicted. The experimental results demonstrated well the effectiveness of the present formulations. The new formula provides an effective approach to implementing a finite-width stress criterion for evaluating the tensile properties of notched fiber-reinforced laminates. In addition, the notch size has a great effect on strength prediction while the fiber direction has a great influence on the fracture mode.

## 1. Introduction

With more and more extensive applications of fiber-reinforced materials, it is common to create different types of openings in composite structures for various purposes in practice. To tackle the mechanical problems associated with structural openings, the characterization of laminates with notches has become a prevalent research topic over the years [1]. Not only does a notch break the continuity of fiber and matrix, but it also tends to produce stress concentration around the notch edge, making a laminated plate more prone to various property losses and even failures. In other words, the mechanical strength of laminates with notches could be substantially reduced [2], thereby decreasing the load bearing capacity of plates. For this reason, the residual strength of notched fiber-reinforced laminates signifies an important topic of research.

No matter whether laminated plates are applied as decorative parts, outsourcing parts, or load bearing components, notch-induced stress concentration presents a common and critical issue in structural analysis [3,4,5,6]. As studied by Nassiraei and Rezadoost, the stress concentration factors (SCFs) in circular hollow-section X-connections retrofitted with fiber-reinforced polymer (FRP) under compressive load have been evaluated [7]. The influence of notch design on failure mode, mechanical behavior, and interlayer damage pattern should be better understood [1,8,9]. In the available literature, most studies on notched laminates can be categorized into the following two types:

The first group focuses on the effects of the number, shape, and size of notches on the mechanical properties of laminates. In this regard, Cunningham et al. [10] studied the influence of notch distribution on the tensile behavior of glass-fiber-reinforced composite plates, in which strain distribution was quantified by the DIC method to identify stress concentration. Xu et al. [11,12] investigated the size effects of sharp and blunt notches on quasi-isotropic laminates; they analyzed the stress concentration around the crack tip by using intermittent testing and X-ray computed tomography (CT). O’Higgins et al. [13] analyzed the effect of specimen size on tensile behavior based upon linear elastic fracture mechanics (LEFMs) and Weibull theory; they found that carbon-fiber-reinforced plastic (CFRP) laminates had higher fracture strength, whilst glass-fiber-reinforced plastic (GFRP) specimens had greater ultimate strain. It should be noted that the progressive damage mechanisms of CFRP and GFRP materials were fairly similar [13,14,15].

The second group focuses on the laminated configuration, such as the lay-up sequence, the lay-up thickness, and the lay-up angle. In this respect, Wisnom and Hallett [16] investigated the effect of plate thickness on the tensile strength and failure mode of quasi-isotropic carbon fiber laminates; they found that delamination plays a key role in the in-plane strength and failure mechanism of notched tensile specimens. Ercin et al. [17] conducted an experimental study on the influence of notch size and lay-up sequence on tensile and compressive behavior; they found that the strength of laminates decreases with an increasing notch size, in which DIC was used to obtain the full-field displacement of the specimen [18,19].

Most of the above studies on the stress concentration of open-hole fiber-reinforced laminates are based upon the Whitney–Nuismer (WN) stress criteria [17,20], in which an infinitely wide plate model has been considered. However, when the ratio of the notch size (2C) to the dimension of an entire part (W) becomes comparable, such a notched plate should not be treated as an “infinitely wide” plate [21]; in real life many parts cannot be regarded as infinite-width plates because of the sizes of holes, so it is imperative to study the strength prediction of finite-width fiber-reinforced materials. For this reason, this paper presents a new stress model for “finite-width” laminates, in which a T stress is introduced to derive novel formulations based upon the WN stress criteria. Not only do the present criteria cover the plates with an infinite width (W/2C≥6) that can be properly predicted by the original WN stress criteria, but they also provide better predictions of the tensile properties of plates with a finite width. According to the experimental results in [22], the predictive results of the WN stress criterion are found to be fairly poor, especially for structures involving finite-width straight slots with large holes [23,24,25]. In this present study, the failure mechanism and residual strength of the laminates with straight notches are explored based upon experimental and analytical methods. Two groups of specimens are prepared to verify the predictive accuracy of the finite-width stress criterion, in which the notch size is taken as a variable. Additionally, the prediction results obtained using the finite-width criterion are compared with those obtained using the WN stress criteria and experimental results to verify the accuracy and applicability of the finite-width criterion. In addition, other structural variables, such as the lay-up sequence, lay-up angle, and lay-up thickness, are considered in the experiments to further validate the applicability of the proposed finite-width stress criterion to fiber-reinforced materials. However, this prediction method is slightly more complicated than the WN stress criteria, and pre-experiments are needed to determine the applicable characteristic length. In addition, since this kind of criterion is based on quasi-static load and homogeneous materials, in addition to having good applicability to materials with quasi-isotropy and load symmetry, only 0°/90° and +45°/−45° symmetric layering angle cases are studied in this paper. In recent years, Mohammad Reza Khosravani et al. [26] studied the fracture behavior of anisotropic materials and A. Sharma et al. [27] studied the dynamic fracture toughness of particulate polymer composites. They studied the above problems via numerical simulations and artificial neural networks, respectively. However, there are many shortcomings in theoretical calculation and prediction; therefore, further studies on the applicable criteria for other layering angles and being under dynamic load are still necessary.

## 2. Theoretical Stress Analysis

### 2.1. Stress Concentration Factor

To investigate the stress concentration and residual strength of notched laminates, this study adopts straight-notched fiber-reinforced composite laminates. The stress concentration coefficient and residual strength are analyzed here to correlate with the experimental and finite element (FE) analysis results.

As shown in Figure 1, the laminated plate specimen has a width, W, and a centrally located transverse straight notch with a length, 2C, subject to a far-field tensile load, ${}^{\infty }$. It is assumed that the y-axis is along the loading direction and that the x-axis is perpendicular to the loading direction, which are, respectively, represented by subscripts 1 and 2 in the following formulae.

Based upon the theory of linear elasticity, the finite-width stress concentration coefficient (SCF), $K_{t}\left(x\right)$, in the x-axis can be expressed as [28]:(1)$$K_{t}\left(x\right)=R_{K}\frac{\sigma^{\infty }}{2}\left\{2+{\left(\frac{C}{x}\right)}^{2}+3{\left(\frac{C}{x}\right)}^{4}-\left(K_{T}^{\infty }-3\right) [5{\left(\frac{C}{x}\right)}^{6}-7{\left(\frac{C}{x}\right)}^{8}]\right\},\;C<x\leq \frac{W}{2}$$
where x is the distance away from the center of the straight-notched (SN) specimen [1]; $K_{T}^{\infty }$ and $R_{K}$ are the stress concentration coefficient and the finite-width correction coefficient, respectively, of the laminate with an infinite width [28]:(2)$$K_{T}^{\infty }=1+{ [\frac{2}{A_{22}}\left(\sqrt{A_{11}A_{22}}-A_{12}+\frac{A_{11}A_{22}-A_{12}^{2}}{2A_{66}}\right)]}^{\frac{1}{2}}$$
(3)$$R_{K}=\frac{K_{T}}{K_{T}^{\infty }}={\left\{\frac{3\left(1-\frac{2C}{W}\right)}{2+{\left(1-\frac{2C}{W}\right)}^{3}}+\frac{1}{2}{\left(\frac{2C}{W}M\right)}^{6}\left(K_{T}^{\infty }-3\right) [1-{\left(\frac{2C}{W}M\right)}^{2}]\right\}}^{-1}$$
where $A_{ij}$ is the plane stiffness matrix. Parameter M can be calculated as follows:(4)$$M^{2}=\frac{\sqrt{1-8 [\frac{3\left(1-\frac{2C}{W}\right)}{2+{\left(1-\frac{2C}{W}\right)}^{3}}-1]}-1}{2\left(\frac{2C}{W}\right)}$$

### 2.2. WN Stress Criteria

Whitney and Nuismer [29] developed two stress fracture models, namely the point stress criterion (PSC) and average stress criterion (ASC), for orthotropic plates containing a circular hole or straight crack to predict the failure of notched plates when W/2C≥6. In PSC, failure occurred when stress in a notched plate at distance $d_{0}$ away from a hole or crack is equal to the strength of a corresponding non-notched plate, as shown in Figure 2a. In ASC, failure is assumed to occur when the average stress over distance $a_{0}$ is equal to the strength of a non-notched plate, as illustrated in Figure 2b. It should be noted that the abovementioned characteristic lengths, $d_{0}$ or $a_{0}$, can be considered to be material properties [30], which are independent of the geometry and stress distribution of specimens. In a laminate with a straight crack, stress ${}_{y}$, ahead of the crack tip, can be approximated as [31,32,33]:
(5)$$\sigma_{y\left(x,0\right)}=\frac{K_{I}x}{\sqrt{\pi C\left(x^{2}-C^{2}\right)}}=\frac{\sigma^{\infty }x}{\sqrt{x^{2}-C^{2}}}\;,\;C<x\leq 1.1C$$
where $K_{I}$ is the stress intensity factor of mode I fracture.

PSC-based failure for a laminate with a straight crack can be determined as:(6)$$\frac{\sigma^{\infty }}{\sigma_{0}}={\left(1-\xi_{3}^{2}\right)}^{\frac{1}{2}}\;\;\;,\;\;\;\xi_{3}=C/\left(C+d_{0}\right)$$

TASC-based failure for a laminate with a straight crack is determined as
(7)$$\frac{\sigma^{\infty }}{\sigma_{0}}={\left(\frac{1-\xi_{4}}{1+\xi_{4}}\right)}^{\frac{1}{2}},\;\xi_{4}=C/\left(C+a_{0}\right)$$

In Equations (6) and (7), it is assumed that $\sigma^{\infty }$ is applied parallel to the y-axis at an infinite distance (far field) of a plate containing a straight notch of length 2C subjected to a uniform stress; $\sigma_{0}$ denotes the failure stress of a corresponding non-notched plate.

### 2.3. Finite-Width Stress Model

While the WN stress criteria can be used to analyze the laminated plate with infinite width, a real structure would more likely present in a notched laminate plate with a finite dimension [34], in which the edge could considerably influence the stress status [35]. Regarding the strength of finite-width notched laminated plates, substantial research efforts have been made to improve the stress criteria in laminated plates with circular holes. For example, high-order coefficients were suggested to make the stress criteria more accurate, so as to reduce the influence of edge stress [31]. Nevertheless, it was found that even if a specimen with a circular opening does not meet the condition of infinite width (W/2C≥6), the error of strength prediction is less than 10%, which is often within an acceptable range in practice [22].

For a straight-notched specimen, however, the experimental strength gradually deviates from an analytical curve with an increasing notch size when W/2C<6, indicating the necessity of further research on laminates with a straight notch [22]. Therefore, new strength prediction stress criteria need to be established for finite-width laminated plates with a straight notch.

When W≫2C (e.g., W/2C≥6), Equations (6) and (7) are considered sufficiently accurate [4,31,36]. In the present experimental tests, W/2C=12, 9, 6, 4.5, 3, 1.5, which involves quasi-infinite and finite-width conditions. In particular, finite width is modeled under traction, T, stress and the equilibrium condition here. On the basis of elasticity theory, ${}_{y}$ at y=0 can be approximated as:(8)$$\sigma_{y\left(x,0\right)}=\frac{\sigma^{\infty }x}{\sqrt{C\left(x^{2}-C^{2}\right)}}+T\;\;,\;C<x\leq \left(W/2\right)$$
where T stands for *T* stress, which can be derived from the equilibrium equation as:(9)$$W\sigma^{\infty }=\overset{\left(W/2\right)}{\underset{a}{\int }}\left(\frac{\sigma^{\infty }x}{\sqrt{C\left(x^{2}-C^{2}\right)}}+T\right)dx\;,\;T=T_{k}\sigma^{\infty }$$
in which parameter $T_{k}$ can be calculated according to the theory of fracture mechanics, as:(10)$$T_{k}=\frac{\left(W/2\right)-\sqrt{{\left(W/2\right)}^{2}-C^{2}}}{\left(W/2\right)-C}$$

By using Equation (8), the modified fracture stress criteria for finite-width laminated plates with a straight crack can be derived as follows:

PSC failure:(11)$$\frac{\sigma_{N}^{\infty }}{\sigma_{0}}=\frac{\sqrt{1-\xi_{3}^{2}}}{1+T_{k}\sqrt{1-\xi_{3}^{2}}}$$

ASC failure:(12)$$\frac{\sigma_{N}^{\infty }}{\sigma_{0}}=\frac{\sqrt{1-\xi_{4}}}{\sqrt{1+\xi_{4}}+T_{k}\sqrt{1-\xi_{4}}}$$

Equation (10) indicates that when W≫2C, $T_{k}$ approaches zero. In this case, T stress can be neglected in Equation (8); in this case, Equations (11) and (12) can be simplified as Equations (6) and (7), respectively. Therefore, the stress failure model of composite laminates with straight slot cracks of finite width proposed here is considered an extension to the original WN stress model.

## 3. Experiment

### 3.1. Materials and Geometry of Specimens

The mechanical properties of laminates strongly rely on the lay-up angle, lay-up sequence, and ply materials [37]. Hence, laminated plate specimens with different lay-ups, notch sizes, and specimen sizes are considered here to scrutinize the credibility of the present stress formulations for finite-width laminates with straight notches.

First, carbon-fiber-reinforced plastic (CFRP) laminates made by VARTM technology are studied here [22,38]. The carbon-fiber-reinforced laminates are hot-pressed by stacking plain woven preimpregnated materials (T300, 3K) supplied by Guangwei Group Co., Ltd. (SD, WEH, CHN). The fiber direction can be adjusted by rotating layers. In this study, 0°/90° layering was mainly used, while +45°/−45° layering was used for minority specimens. The composite laminates used for subsequent processing need to ensure sufficient dimensions after cutting the 15 mm rough edges during the process. To prevent defects and keep a smooth surface (to avoid influential speckle treatment for DIC measurements) [18], a CNC milling machine is used to cut specimens from the fabricated blank plates. Since this study aimed to compare a theoretical model with the experimental results of specimens with straight notch, a sharp milling cutter with a diameter of 1 mm was used. After cutting a certain number of notches in specimens, a new milling cutter needs to be installed to ensure the consistent quality of the notches without cutting damage in the edge.

As shown in Figure 3a, the size of specimens is 250×36 mm; the total thickness of 8-ply composite laminates is 2±0.4 mm; and the length of a standard straight notch is 6±0.06 mm (W/2C = 6, right on the threshold between infinite and finite width) in line with ASTM D5766 [30]. The clamping end is 50 mm in length and is retained on both sides of the specimen. A straight notch specimen with a crack tip formed due to the limitation of processing technology (as shown in Figure 3a).

To facilitate the analysis, the labeling system used in [22] was adopted in this study as well; some new letters were added to represent the new specimens. Taking CF-SN-3 as an example, CF stands for carbon-fiber-reinforced composite, SN represents a straight notch, and 3 presents the notch length in the specimen. In the specimens CF-SN-S-3, CF-SN-M-4, and CF-SN-L-8, CF and SN are the same as above; S, M, and L represent the adjustments of specimen size (with the same value of W/2C as the standard specimen). Among them, the variables of different lay-up sequence groups and different laminated thickness groups are fiber direction and lay-up thickness, so the hole size is the same as that of standard specimens (6 mm). An equal ratio of the W/2C group maintained the same size ratio of W/2C as the standard specimens, but due to the size limitations of the stretching machine the width of the largest part was 48 mm, and the size of the other two specimens was half that of the largest specimen and the standard specimen, respectively. The overall size of all specimens in the equal specimen size group was the same (250 mm × 36 mm). The opening size was set as a geometric sequence (6 mm, 12 mm, and 24 mm), and a specimen (2C = 3 mm) that could be regarded as infinite-width was set as a control specimen.

Table 1 summarizes the specific size of each specimen. Note that numbers 3, 4, and 8 at the end of each sampling label indicate the notch size (in mm). In the experimental group with the different layer angles, letter A represents the normal woven fabric orthogonal lay-up, and letter B represents the 45°-oriented lay-up of the plate mold. To denote the laminated thickness, labels ply12 and ply16 are used to distinguish the thickness of the non-8-ply specimens. For example, ply12 indicates 12 layers of woven fabric (the standard specimen is 8 layers) in one specimen, and each layer has a thickness of 0.25 mm; thus, the thickness of this specimen is 3 mm. In [0°/90°]_8_, [0°/90°] represents the lay-up Angle, the following 8 represents the number of the front lay-up, and s in [(0°/90°)/±45°]_2S_ represents the symmetric copy of the front lay-up. The specific specimen types are shown in Figure 3. Each group of specimens was repeatedly tested 3 times.

### 3.2. Testing

As shown in Figure 4, a fatigue tensile testing machine INSTRON8801(INSTRON, SH, CHN) was employed for the experiments. The displacement control mode was used here at a speed of 2 mm/min (based upon the ASTM standard [30,39]). The ambient temperature was set to be 25 °C and the humidity to be 33% RH.

To ensure the measurement quality of digital image correlation (DIC), one surface light source was placed on each side of the testing machine [40]. Full-field surface strain was measured in real time using the ARAMIS v6.3.1 (GOM mbH, BWE, GER). In this study, the pixels of the CCD (charge-coupled device) camera were set to be 2248×2050, and the focal length was 50 mm. The DIC technique was also used to evaluate the local strain concentration on the surface and any damage development in the region of interest [41,42,43]. In order to better implement the DIC measurement on the specimens, the DIC cameras were configured to cover the entire sample area. In this study, non-uniform strain gradients may only develop in a small area near the opening; thus, a sufficient resolution needs to be ensured.

## 4. Results and Discussion

### 4.1. Load–Displacement Curves

The graphs presented in this section show the load–displacement curves for each group of specimens. As shown in Figure 5a, the notch sizes to a certain extent affect the stiffness of the specimens, as discussed in [22]. It can be observed that when the crack length is less than or equal to 6 mm, the effect of notch size on stiffness can be neglected as the specimen can be approximately treated as an infinite plate. However, when the notch size is greater than 6 mm the effect of edge should not be ignored, where the stiffness is seen to decrease to a certain extent.

As shown in Figure 5b, the label of testing specimens indicates that the specimen size gradually expands with an increasing notch size to retain the same size ratio, such that the size effect can be ignored. The ratio between the cross-sectional areas of the joints on both sides of their notches is the same as that of non-notched plates with the same size. The larger the plate size, the higher the peak load. The slopes of the load–displacement curves obtained by the group of specimens in Figure 5b are fairly similar, except for CF-SN-M-4, which might be because of the error caused by processing the small-sized specimen with the same-sized cutter. It indicated that the specimens in this group are not substantially affected by the size when ignoring CF-SN-M-4, and they can be used as a representative group to verify the accuracy of the proposed stress criterion formula.

For the specimens with different lay-ups, as shown in Figure 6a, the fiber direction (ply angle) exhibits a significant effect on the tensile strength, whereas the lay-up sequence does not. Neither of the specimens with ±45° layers have shown superiority in terms of the tensile strength; their tensile strengths are slightly lower than that of the orthogonal counterpart (0°/90° woven layer), whilst their toughness and ductility are slightly higher than those of the orthogonal counterpart. This is mainly because the orthogonal specimen used 8-ply, which has the best tensile properties in the directions of 0° and 90°; therefore, its tensile strength is higher than that of the other specimens. In the other two types of specimens (CF-SB-[ABAB]s and CF-SB-[AABB]s), four pieces of braided fabric layers in ±45° were added to the other four 0°/90° layers. The results showed that the tensile strengths of these two different braided fabric specimens decreased with increasing failure displacement.

Figure 6b plots the force–displacement curves for specimens with different layer thicknesses. Apparently, laminated thickness significantly affects the tensile strength. The greater the thickness the higher the peak tensile load, which bears similarity to the non-notched specimens. Previous studies have shown that specimens with a greater thickness are more likely to be affected by stratification in tensile tests [17]. In this study, specimens with a thickness of 4 mm (CF-SN-ply16) all exhibited certain curve bending in the elastic stage, which is mainly associated with a stratification-induced reduction in tensile strength. Although the ultimate failure of this group of specimens was still fracturing in the fiber and matrix, local delamination still occurred during the loading process.

In addition, the load–displacement curves of these two groups of specimens in Figure 6a were basically identical, revealing that the tensile properties are less associated with the lay-up sequence and were only related to the number of layers and ply angle.

### 4.2. Comparison of Average Failure Strain

After post-processing the DIC data, the strain contours were obtained. The strain was analyzed for the same specimens with different openings, as well as for different specimens with different opening sizes, to characterize the mechanical properties. Figure 7 compares the mean failure strain of the two groups of straight-notched specimens. In the left part of the graph (Figure 7a), the notch size increases gradually, while the specimen size remains the same. It can be seen that with an increase in the opening size, the strain concentration at the notch edge increases, and the average failure strain decreases. In the right part of the figure (Figure 7b), the size ratio of specimens, W/2C, is kept the same as the standard specimen. With increasing crack size, the change in the average failure strain does not seem significant except for CF-SN-S-3. This means that the strain concentration at the crack tip does not increase linearly. This difference provides data for verifying the proposed stress criterion for its suitability to different cases.

Figure 8 compares the average failure strains for different specimen thicknesses and lay-up sequences. It can be seen that the average failure strain of specimen CF-SN-6 (2 mm in thickness and eight layers in lamination) is fairly close to that of specimens with 12 layers and 16 layers (CF-SN-ply12 and CF-SN-ply16, respectively). Thus, the lamination thickness influences only the tensile strength, not so much the stress concentration in the specimen notch. On the contrary, although ply sequences do not have much of an effect on tensile strength, the fiber direction exhibits an evident effect on the strain concentration of the notch.

Combining the stress–strain curves in Figure 9 and the average strain diagram in Figure 7, it can be found that the stress concentration at the crack tip of specimen CF-SN-[ABAB]s is more evident than that of specimen CF-SN-[AABB]s (i.e., the average strain of CF-SN-[ABAB]s is higher). By referring to the average strain of the CF-SN-6 specimen, it can be seen that the degree of strain concentration is relatively small for the orthogonal fabric. The orthogonal layer and ±45° fiber overlay of an [ABAB]s-type specimen play a key role in bearing mechanical loading. Meanwhile, in [AABB]s-type specimens, the outer four layers (0°/90°) bear stress first, and the inner ±45° fiber layers experienced cracking after being stretched. For this reason, the fracture strain detected by DIC on the outer surface is relatively small.

### 4.3. Characteristics of Notched Fiber-Reinforced Specimens

Table 2 summarizes the ratio of the peak tensile load to the tensile strength of non-notched counterparts with different notch sizes, different lay-up sequences, and different sample thicknesses. The tensile strength ratio quantifies the tensile strength of the notched specimen with respect to the tensile strength of the intact specimen with the same overall size. The notch sensitivity signifies another parameter that represents the change trend in the strength of the material due to the presence of a notch; its value is inversely proportional to the tensile strength ratio [44]. In general, the greater the notch sensitivity the lower the strength of the material [45]. With an increasing notch size the notch sensitivities of both the equal ratio (W/2C) group and the equal size group increase, indicating that the notch size affects the strength of the laminates. When the notch size increases from 3 to 6 mm, the notch sensitivity of the equal size plate and equal ratio plate increases by 20.4% and 26.4%, respectively. This means that the size of the specimen considerably affects the stress state of the specimen.

When comparing the notch sensitivities of the CF-SN-[AABB]s, CF-SN-[ABAB]s, and standard specimens, the type of lay-up exhibits noticeable influence on the stress state in the specimens, while the lay-up sequence has relatively little influence. It can be seen that, for straight-notched laminates, the 45° lay-up makes them have a higher strength ratio. From comparison of the 12-layer, 16-layer and standard (8-layer) specimens, it can be seen that the notch sensitivity increases with the number of layers.

### 4.4. Analysis of Progressive Failure through Strain Distribution

Figure 10 shows that the DIC strain field of different open-hole specimens with the same specimen size (W) corresponds to Figure 5a and Figure 7a under a 99% tensile load. It is apparently the case that the strain contours are all concentrated around the two end tips of the straight crack. From the figure, it can be seen that the strain concentration area extends in the region with a large aperture towards the left/right edge of the specimen, which explains why the discrepancy between the experimental data and the theoretical values becomes fairly large where the specimen cannot be regarded to be infinite.

Figure 11 compares the strain fields of the equal ratio (W/2C) specimens, corresponding to Figure 5b and Figure 7b, under a 99% tensile load. The strains in the figure are all concentrated around the two end tips of the straight notch crack. Due to the concurrent enlargement of specimen size in this case, the strain concentration areas do not diffuse to the edge. Additionally, the strain contours are largely consistent after being scaled to the same size. Therefore, to avoid the ineffectiveness of the “infinite-width” condition due to the oversizing of a hole, the equal ratio magnification of specimens can, to a certain extent, ensure predictive accuracy. However, as can be seen from the right two graphs of Figure 11, due to the size of the cutter the size of the straight notches along the tensile direction is unchanged. Therefore, with the reduction in the notch size, the straight slot hole gradually approaches the rectangular notch, and the machining error of the tool may also have a greater impact on the test data. Therefore, the difference between the curve trend of CF-SN-M-4 in Figure 5b and the other three types of specimens is probably caused by the error generated during processing. Therefore, to avoid this situation next time, we can do the following: 1. Increase the number of repeated tests of specimens with a small notch size and eliminate the differentiation items. 2. Use a smaller cutter. 3. Try to avoid testing specimens with too small a notch size; for example, add a test group to the direction of a large size. However, because the size of the tensile machine limits the upper limit of specimen size, more factors need to be considered in this scheme.

Figure 12 shows the strain field of specimens CF-SN-[AABB]s and CF-SN-[ABAB]s under a 99% tensile load. A rectangular coordinate system is established for discussion here. Assume that the loading direction is y and that the corresponding perpendicular direction is x, and that the strain in each direction can be displayed by a color contour. As seen in Figure 12a,b, the strain concentrated in the areas around the tips of the straight notch. It can clearly be seen that the longitudinal strain field’s, $\varepsilon_{yy}$, upper limit, as well as the difference between the upper limit and lower limit of the CF-SN-[ABAB]s specimen, are greater, meaning that the strain concentration and the failure strain are more manifest. By comparing the strain concentration areas, it is clear that the strain in the axial direction cannot be ignored. In addition, the contour values of shear strain were distributed alternately in positive and negative areas, as shown in strain field $\varepsilon_{xy}$.

In conclusion, strain concentration appears around the crack tips of the straight notch when the specimen was stretched; the stress concentration area takes a cone shape and diffuses outward parallel to the crack. A finite-width plate can determine whether the stress concentration area can diffuse to the specimen boundary under peak loading, which will directly affect the prediction accuracy of the WN stress criteria. Therefore, the inaccurate prediction induced by a large notch size could, to a certain extent, be alleviated by expanding the specimen size in a form of an equal ratio to the increasing notch size.

### 4.5. Failure Characteristic Analysis

In order to scrutinize the fracture details for analyzing the failure mode in different scales, JMS-IT300 scanning electron microscopy (SEM) was used to magnify the failure site 65, 550, and 4000 times; the corresponding images are displayed in Figure 13, Figure 14 and Figure 15. By combining such images with those presented Figure 16 and Figure 17, the failure modes of each sample can be analyzed from the viewpoints of fiber, fiber bundles, and the overall levels of the specimens.

As can be seen from Figure 13 and Figure 14, there is no significant difference in the failure modes for all of the specimens composed of 0°/90° fiber angles with the same thickness. Failure modes present mostly in fiber fracture and a small part of fiber pull-out and delamination, with a relatively smooth fiber fracture face. It can be seen from Figure 15 that the sample containing a 45° fiber orientation was pulled out and that the lay-up was delaminated severely; in particular, the specimen with a 45° lay-up (CF-SN-[ABAB]s) was stratified extensively, where the fiber fracture face is fairly rough. This is because of the fact that it can effectively prevent the propagation of the crack when the direction of the fiber is orthogonal to the direction of the crack. For specimens with a different thickness, in particular CF-SN-ply16, small pieces of broken fiber can be seen in both 550× and 4000× images. It is thus speculated that the overall strength and failure load increase with increasing laminated thickness, which leads to fiber rupture under greater mutual extrusion pressure before complete failure.

As shown in Figure 17, the outer plain fabric of the CF-SN-[AABB]s specimen was found to be cracked first; the internal ±45° layer failed before whole specimen was completely broken, indicating that the specimen first cracked at the crack tip of the outer four-layer (0°/90°) braid along the x direction. The cracks first appeared in the gap between the two adjacent fiber bundles; however, the fiber bundles of the middle 4 layers (45° overlay) do not take loads directly in the fiber direction, leading to frictional contact, deformation, and even sliding between the fiber bundles, which will be pulled and broken only when the outer layer fails. After failure, the testing machine began unloading the specimen, such that the inner fiber bundle may not be broken completely. In contrast, the CF-SN-[ABAB]s specimen was fractured almost completely, mainly because the laminated plates have an orthogonal lay-up configuration of ±45° between neighboring layers; stress is more evenly distributed, and fracturing occurs almost at the same time in every two layers (including the outermost layer), such that the failure of the specimen finally led to a debris morphology closer to that of the pure orthogonal layer of the specimen.

In conclusion, the fracture mode of the specimen largely relied on the lay-up angle and layer thickness, while the notch size has a less significant effect on the failure mode. In addition, fiber pull-out mainly occurred in the specimen containing only 0°/90° fiber and a relatively thin thickness, where the fiber fracture face is relatively smooth. As the specimen becomes thicker, the tensile strength increases, the mutual extrusion pressure inside the fiber becomes stronger, more crushing occurred inside the fiber, and the fiber fracture face became fairly rough. Due to the non-uniform load bearing capacity by differently oriented fibers in the adjacent layer, the main failure mode of mixed laminated specimens with a ±45° layer is delamination.

## 5. Characteristic Length and Predictive Models

Point stress criterion (PSC) and average stress criterion (ASC) can be validated by experimental data [22], which have exhibited that when the notch size of a specimen meets the condition W/2C≥6 the specimen can be approximated to be an infinite-width plate, and these two stress criteria have a higher prediction accuracy. Whereas when the size of the notched specimen is W/2C<6 the specimen is presented in a finite-width, the predictive values of the two stress criteria gradually exhibit a certain discrepancy. This becomes more evident when the notched shape is a straight slot. In order to explain this situation more clearly and tackle the strength prediction problem in a finite-width plate with a straight notch, a strength prediction model for such laminates has been proposed in Section 2.3, as above. This section will first investigate whether the characteristic length can be regarded as a material property. Then, the WN damage-based prediction accuracy for specimens with a large straight notch is analyzed in Section 5.2. Finally, the predictive error of the newly proposed formula is verified in Section 5.3.

### 5.1. Calculation of Characteristic Length

Combining the strength ratio calculated by the experiments of non-notched specimens and corresponding notched specimens, the characteristic lengths, $d_{0}$ and $a_{0}$, of these two groups can be obtained by using Equations (6) and (7) of the WN stress criteria or Equations (11) and (12) of the finite-width stress criteria, respectively. As mentioned in [29,46], the characteristic length can be regarded as a material property, which is independent of the notch size. This section will verify the solutions of these two sets of formulae.

Figure 18 and Figure 19 compare the characteristic lengths for the experimental group with the equal specimen size and the experimental group with the equal size ratio, respectively. It can be seen that, for the equal size ratio group (Figure 19), the size ratio of the notch to the specimen remained at 1:6 (which can be approximately regarded as an infinite-width plate), so the characteristic length of each specimen in this group is fairly consistent. However, the characteristic lengths in the equal specimen size group (Figure 18) that were calculated under the WN-stress-criteria-based formula vary greatly, especially in a greater notch size, which cannot be regarded as an infinite plate.

For further analysis, the standard deviation and coefficient of variation (Cov) of the characteristic lengths of these two groups were calculated, as listed in Table 3. It can be seen that the characteristic lengths obtained from these two methods have a fairly small coefficient of variation for the equal ratio group. For the equal size group, the original great coefficient of variation was reduced by nearly half, less than 15%, after the use of the proposed finite-width stress criterion. Therefore, the characteristic length can be approximately regarded as a constant in this situation.

The above analysis indicated that the characteristic lengths, $d_{0}$ and $a_{0}$, can be approximated as constants when the specimen is regard as an infinite-width plate that well satisfies the WN stress criteria. Thus, such lengths can be used to characterize material behavior that is independent of the geometric shape and stress distribution in the specimen. Even if the plate is not really infinite, the values of the characteristic lengths, $d_{0}$ and $a_{0}$, can still be approximately regarded as constants under the proposed formula for finite-width stress criteria. Meanwhile, the wide applicability of the finite-width stress criteria for a specimen with a large notch size can be further justified. It is worth mentioning that the applicability of the point stress criterion is better than that of the average stress criterion in terms of the stability of characteristic length, which can be seen in Table 3.

In [22], the values of $d_{0}$ and $a_{0}$ are actually the means calculated by least-square fitting in order to achieve a better prediction. To justify the prediction accuracy provided by the new formula, the $d_{0}$ and $a_{0}$ are selected from the actual values calculated by the unperforated specimen with the standard specimen. Figure 20 shows the characteristic lengths of each group calculated from the WN stress criteria. Figure 21 shows the characteristic lengths of each group calculated from the proposed stress criteria for a finite width. It can be seen that both the lay-up sequence and layer number affect the characteristic length; due to the insufficient number of experimental groups and limitations of the resource, more extensive experimental analyses are still required for comprehensive comparisons.

### 5.2. Prediction Based upon WN Stress Criteria

In line with ASTM D 5766, a standard specimen of 250×36 mm has a notch size of 6 mm. Figure 22 compares the experimental value with the predicted value of the equal size ratio group. As per the standard, the size of the specimen needs to be scaled-up when the notch size increases. It was found that the errors of the predicted values were all within 5%, indicating that the WN damage model has good prediction accuracy in regard to the strength of the notched specimens with an infinite width. It is apparently the case that the strength of infinite-width laminates $\left(W/2C\geq 6\right)$ is closer to the average stress criterion curve, whereas the strength of the specimens with a larger notch size (W/2C<6) is closer to the point stress criterion curve, which is consistent with the analysis in the previous section.

In order to more clearly observe the accuracy of strength prediction for finite-width cases, specimens with notch sizes of 12 and 24 mm were added to the experiment. It can be seen from Figure 23 that for a finite-width laminated plate, the prediction from the WN stress criteria exhibits a significant discrepancy from the experimental results. When the notch size is 12 mm, the predicted error from the point stress criterion is about 9.1% and the average stress criterion is about 11%. When the notch size is 24 mm, the predicted errors from the point stress and average stress are 27% and 29.5%, respectively, which are far greater than the accepted level for such an experimental study. Therefore, it is not suitable to use the WN stress criteria for predicting the strength of finite-width notch laminates. Figure 24 presents a bar chart to more clearly compare finite-width results in terms of the stress criteria.

### 5.3. Prediction Based upon Finite-Width Stress Criteria

Figure 25 compares the results obtained from the new formula with the experimental value for the equal ratio specimens (CF-SN-S-3/M-4/L-8). The results indicate that the proposed new formula for finite-width notched laminates is also applicable to the specimens with infinite-width counterparts. As the T stress parameter, $T_{k\;}$, in the specimen changes with the notch size of the specimen, different notch sizes cannot be represented by the same theoretical curve. As shown in Figure 26, theoretical prediction and experimental data are compared for the same specimen but different notch sizes. For the notched specimen with W/2C≤6, the finite-width stress criterion can be used with a predictive error of no more than 5%. For the specimens with W/2C≥6, when the notch size is 12 mm the predictive error from the point stress is about 2.6%, and the error between the predicted value of the average stress criterion of finite width and the experimental value is about 4.5%. When the notch size is 24 mm, the predictive errors from the point stress and average stress are 6.7% and 9.5%, respectively. Compared with the original stress criterion, the predicted value of the new formula is closer to the real experimental values, and the error level is less than 10%, indicating that the new finite-width stress criterion formula has a higher predictive accuracy. Similarly, the use of the proposed finite-width formula does not change the trend in experimental values. It is also noted that the strength value of the specimens with an infinite width of W/2C≥6 is closer to the average stress criterion curve, and the strength value of the specimens with a notch size of W/2C≤6 is closer to the point stress criterion curve.

By comparing Figure 24 with Figure 26, it can be seen that, for the two specimens (W/2C>6) with a small notch size, the predictive errors for both stress criteria are small. However, for the last two specimens (W/2C≤6) with a large notch size, the predictive error from the WN stress criteria is clearly greater than those from the proposed finite-width stress criteria. For both of the stress criteria the prediction results from the point stress criterion are better than those from the average stress criterion when the notch size is substantial.

### 5.4. Reliability of the Finite-Width Stress Criterion and Discussion

For straight-notched specimens prepared in-line with the experimental standard, the proposed formula can effectively predict the tensile strength of notched specimens with both an infinite width and finite width. To verify the general applicability of the new formula, several further experiments were considered in this study. Two experimental specimens in the first group, CF-SN-[ABAB]s and CF-SN-[AABB]s, were prepared according to different lay-down hierarchies (fiber direction). Furthermore, CF-SN-ply12 and CF-SN-ply16, with lay-down thicknesses of 3 mm and 4 mm, respectively, were in the second group. As shown in Section 5.1, the predicted values largely depend on the characteristic length; the two groups of experiments mentioned above have two kinds of control specimen (CF-SN-[ABAB]s, CF-SN-[AABB]s) but no standard specimen. Therefore, no matter which kind of experiment is selected as the standard specimen, it is inappropriate to calculate the characteristic length. Therefore, the characteristic length directly derived from the test values of these two kinds of specimen can be compared with the characteristic length obtained from the test results of the first two groups. As shown in Figure 27, it can be seen that, compared with the standard specimen (CF-SN-6), regardless of adding the 45° layer or changing the laminated thickness, the characteristic length will be affected. Comparing the characteristic lengths of the CF-SN-[ABAB]s specimen with that of the CF-SN-[AABB]s specimen, the lay-up sequence was found to have a limited influence on the characteristic length; however, the specific situation could not be further analyzed due to a lack of control specimens. To sum up, the finite-width stress exhibits good applicability only in predicting straight-notched specimens of the same class (same materials, same lay-up type, and same lay-up thickness).

In order to further illustrate the influence of the lay-up angle on the tensile properties of specimens, Figure 28 compares the load–displacement curves of the CF-SN-B specimen (±45° lay-up, 2C=6 mm) with the original CF-SN-6 specimen. In the initial stage of the load–displacement curve, there is a short linear relationship; thereafter, there is a fairly long nonlinear relationship. It is noted that, with increasing load, the material stiffness gradually decreases until failure. Compared with the non-notched ±45° laminated specimens (CF-SN-B-1,2,3), the load–displacement curve follows the same trend prior to failure. However, the load–displacement curves of different non-notched specimens are quite different when the specimens were loaded to more than 95% of the peak load. This is because, with increasing load, the subtle differences between different specimens become more evident under the non-axial tensile load along the fiber. Thus, the damage difference in each local area of the internal fiber and matrix (including fiber bundle fracture, internal matrix cracking, crack extension, etc.) is further enlarged until one of them breaks first, which leads to different failure cases in different specimens. Therefore, many different failure types are presented throughout these specimens (the resin base falling off, the delamination of laminates, and the fracturing of fiber bundles aggravated), and the specimens failed completely.

It has been shown that the braid direction and laminating direction of the fiber tow of the composite material largely influenced the tensile properties and failure modes of the composite material. Lamination along the direction of ±45° fibers also exhibited a considerable effect on the prediction of tensile strength. The proposed finite-width stress criterion is applicable to the material conforming to the orthogonal fiber distribution, whereas plasticity may need to be considered for other lay-up angles (fiber directions).

It should be noted that, in practical applications, there are various structural openings, and this present study only explored laminated plate specimens with central notches. Non-central notched plates are also widely used in practice, which require further research. In addition, for notches undergoing repair, the residual performance of repaired specimens is also an important topic. In addition, for other common polymer special fiber-reinforced laminates (such as ultra-high-molecular-weight polyethylene fiber and aramid-reinforced bulletproof inserts), notch failure model research still needs to be improved, and it is also an important topic with which to explore the strengthening mechanisms of polymer fibers. To this end, the following studies can be carried out in the next stage of research:(1)Explore the mechanical behavior of the notch failure of other polymer fiber laminates and improve the theoretical prediction model.(2)Explore the mechanical properties of non-orthogonal composite laminates and improve the theoretical prediction model.(3)Investigate the mechanical properties of non-central notched composite laminates and establish a suitable theoretical model.(4)Study the patch repair methods, patch types, and mechanical properties of the patched composite laminates.

## 6. Conclusions

In this study, a series of experimental and theoretical analyses have been carried out based upon the stress criteria of notched finite-width fiber-reinforced laminates; the predictive accuracy and credibility of new formulae were verified. The stress concentration of the notch edge and the damage behaviors of different specimens were explored in detail. With limitations, the following conclusions can be drawn:(1)The quasi-static tensile behavior of carbon-fiber-reinforced plastic (CFRP) composites knitted with plain weave depend on the fiber strength in two weaving directions. The failure modes presented in the test are fairly close to brittle fracturing.(2)The fracture behavior of the specimen relies on the lay-up angle (fiber direction) and layer thickness, while the notch size has a less significant effect on the failure mode.(3)The finite-width stress criteria are found to be suitable for notched fiber-reinforced laminate specimens with both an infinite width and a finite width, and the size effect of the specimen almost does not affect the prediction results of the tensile strength when using the proposed formulations.(4)The finite-width stress criteria exhibited fairly good applicability. They can accurately predict the strength of the same kinds of specimens (e.g., the same materials, same lay-up type, and same lay-up thickness) after testing the standard specimen (the specimen used to decide the characteristic length). It can be well-applied to all kinds of fiber rei-forced composites and has great significance for the study of the strength of fiber rei-forced composites.

All in all, the finite width stress criteria substantially extend the prediction range of notched fiber-reinforced materials; however, the applicability of this theory is still worth further research, such as whether it is still suitable for other kinds of fiber-reinforced materials or, in the case of special-shaped notches, multi-notch and asymmetry situations. In addition, the influence of fiber direction on the overall strength prediction is also worth further research. In the future, the artificial neural network method may also be introduced to further expand the scope and accuracy of theoretical prediction.

## Figures and Tables

**Figure 1 polymers-14-05552-f001:**
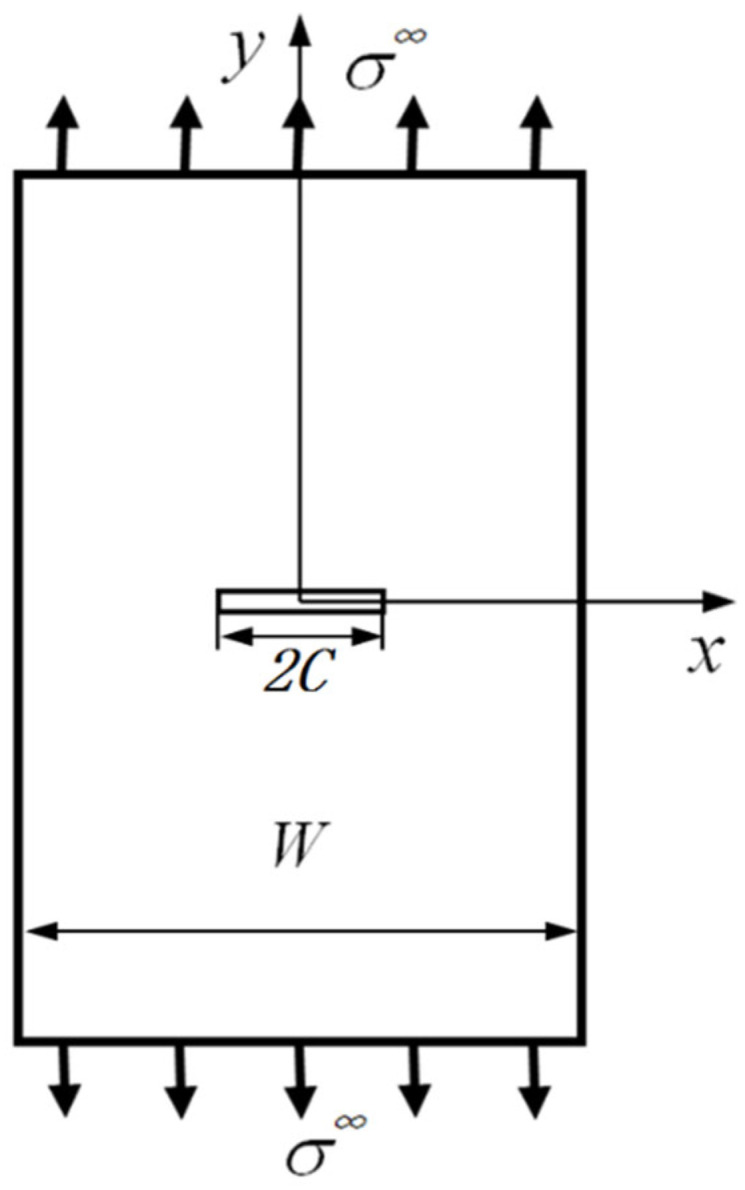
Coordinate system establishment and dimensioning of straight-notched laminated plate under tensile load, where W stands for the width of the specimen and the length of the straight notch is 2C; the direction of the tensile load is along the y -axis.

**Figure 2 polymers-14-05552-f002:**
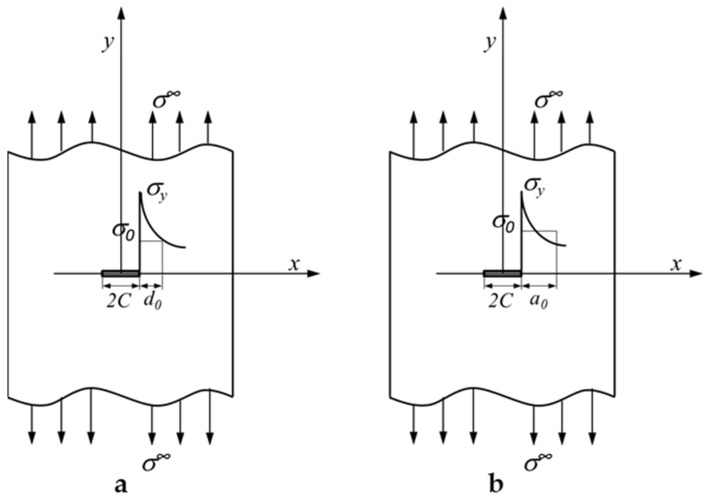
Stress relationship of an infinite-width plate with a straight slot under tensile load under the Whitney and Nuismer criteria. (**a**) Point stress criterion (PSC) failure occurred when stress in the notched plate at distance $d_{0}$ away from a hole or crack is equal to the strength of a corresponding non-notched plate. (**b**) Average stress criterion (ASC) failure is assumed to occur when the average stress over distance $a_{0}$ is equal to the strength of a non-notched plate.

**Figure 3 polymers-14-05552-f003:**
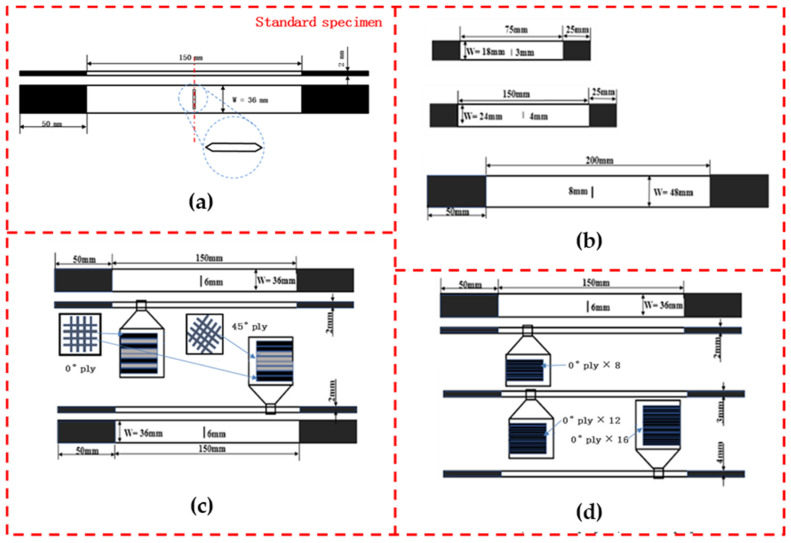
Specimen and configuration. (**a**) Equal specimen size group; (**b**) equal ratio of W/2C group; (**c**) different lay-up sequence group; and (**d**) different laminated thickness group. CF stands for carbon-fiber-reinforced composite; SN represents a straight notch; the following number stands for the notch size; letter A represents the normal woven fabric orthogonal lay-up; and letter B represents the 45°-oriented lay-up of the plate mold.

**Figure 4 polymers-14-05552-f004:**
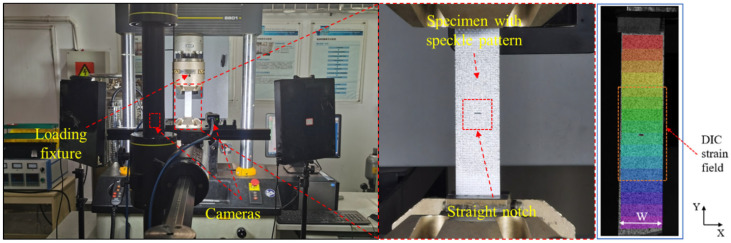
Tensile testing setup, using a fatigue tensile testing machine (INSTRON8801) for the tensile test. Speckle treatment was performed with white and black matte paint to facilitate observation of the experimental process using DIC technology.

**Figure 5 polymers-14-05552-f005:**
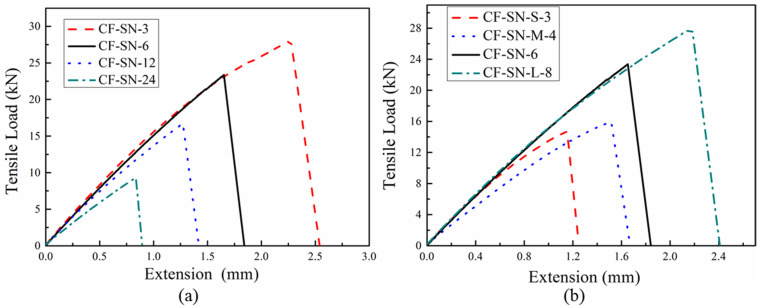
Load–displacement curves of straight-notched specimens under tensile load. (**a**) Equal specimen size group; when the crack length is less than or equal to 6 mm the effect of notch size on stiffness can be neglected. (**b**) Equal size ratio ($W/\left(2C\right)$) group; the stiffness of specimens in this group are not substantially affected by the size.

**Figure 6 polymers-14-05552-f006:**
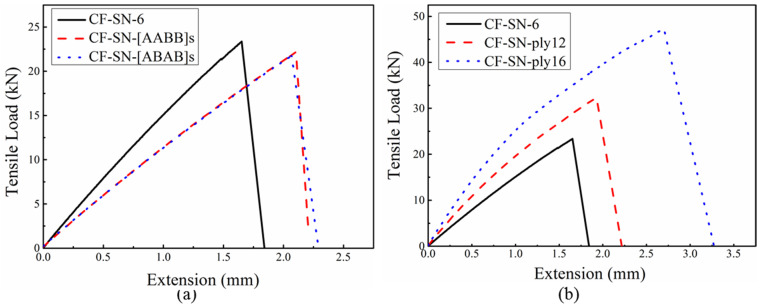
Tensile load–displacement curves of straight-notched specimens under tensile load. (**a**) Different lay-up sequence group; the ply angle exhibits a significant effect on the tensile strength, whereas the lay-up sequence does not. (**b**) Different laminated thickness group; the greater the thickness, the higher the stiffness and peak tensile load.

**Figure 7 polymers-14-05552-f007:**
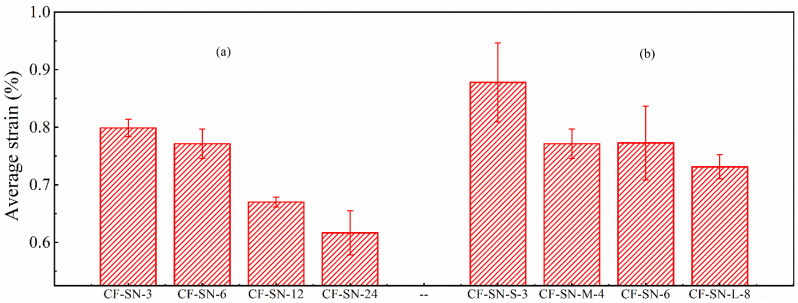
Comparison of the average strain of specimens with different notch sizes. (**a**) Different notch size for the same specimen size. (**b**) Different notch size for the same size ratio, W/2C; the strain concentration at the crack tip does not increase linearly.

**Figure 8 polymers-14-05552-f008:**
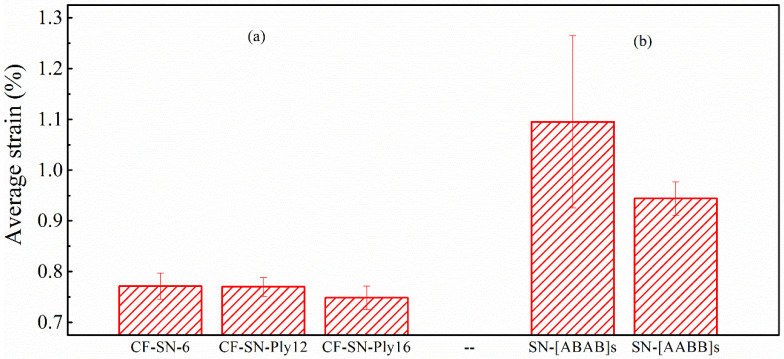
Comparison of the average strain of specimens with different lay-ups. (**a**) Different layer thickness group. (**b**) Different layer sequence group. The degree of strain concentration is relatively small for the orthogonal fabric; the stress concentration at the crack tip of specimen CF-SN-[ABAB]s is more evident than that of specimen CF-SN-[AABB]s (i.e., the average strain of CF-SN-[ABAB]s is higher).

**Figure 9 polymers-14-05552-f009:**
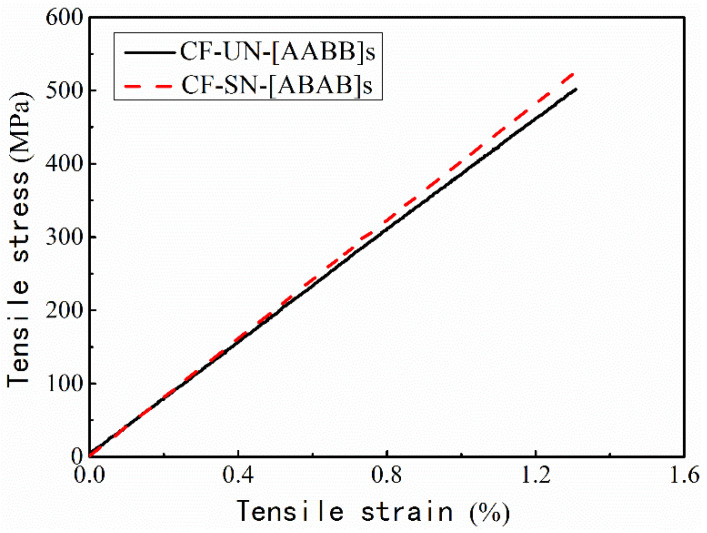
Comparison of the stress–strain relations of specimens with different layer sequences. The curves are almost the same, but the stress concentration at the crack tip of specimen CF-SN-[ABAB]s is more evident than that of specimen CF-SN-[AABB]s.

**Figure 10 polymers-14-05552-f010:**
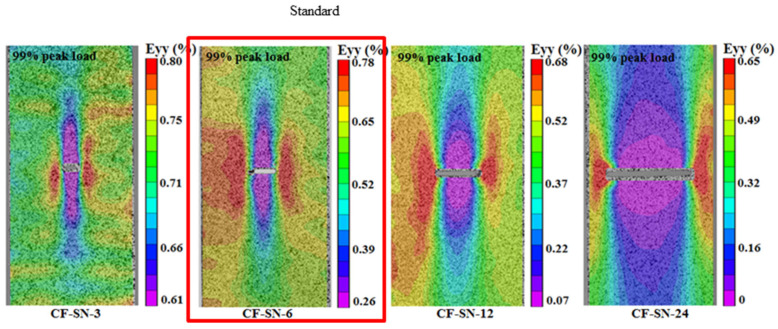
Strain field distribution of straight-notched specimens with the same specimen size (W) under a 99% ultimate tensile load; the strain concentration area extends in the specimen with a large notch towards the left/right edge of the specimen.

**Figure 11 polymers-14-05552-f011:**
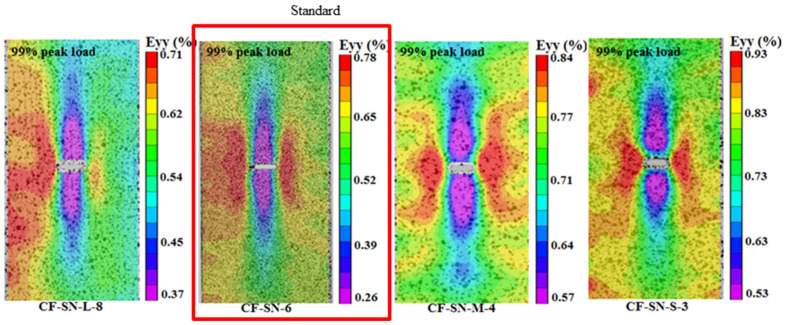
Strain field distribution of straight-notched specimens with the same ratio of W/2C under a 99% ultimate tensile load; the strain concentration areas do not diffuse to the edge due to the concurrent enlargement of specimen size.

**Figure 12 polymers-14-05552-f012:**
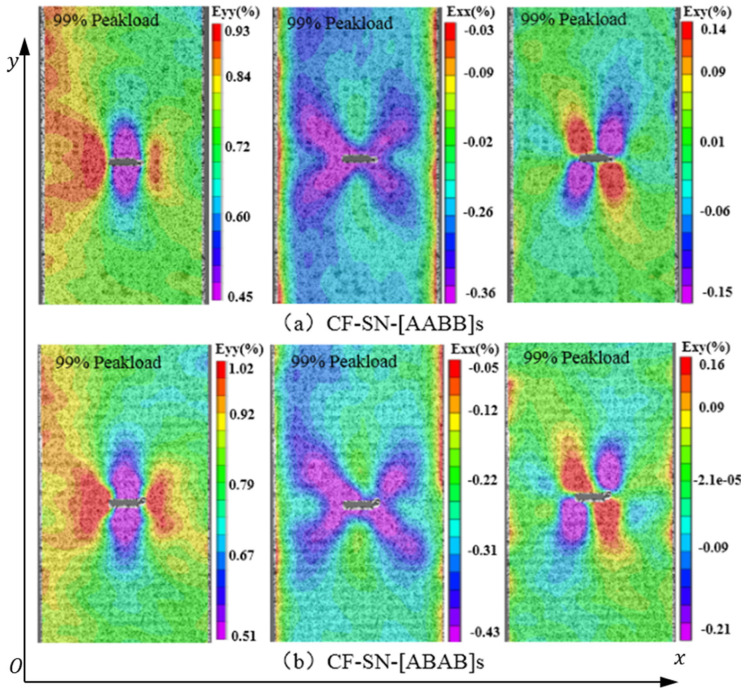
Strain field distribution of straight-notched specimens with different layer sequences under a 99% ultimate tensile load; the strain concentrated in the areas around the tips of the straight notch, and the contour values of shear strain were distributed alternately in positive and negative areas (**a**) CF-SN-[AABB]s, (**b**) CF-SN-[ABAB]s.

**Figure 13 polymers-14-05552-f013:**
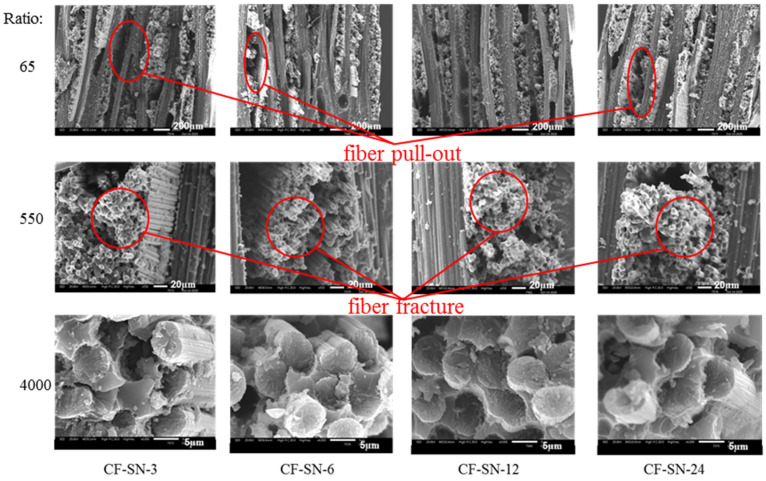
SEM images of the fracture position of the specimen group with an equal size; no significant difference in the failure modes for the specimens of this group. Failure modes present mostly in fiber fracture and a small part of fiber pull-out and delamination, with a relatively smooth fiber fracture face.

**Figure 14 polymers-14-05552-f014:**
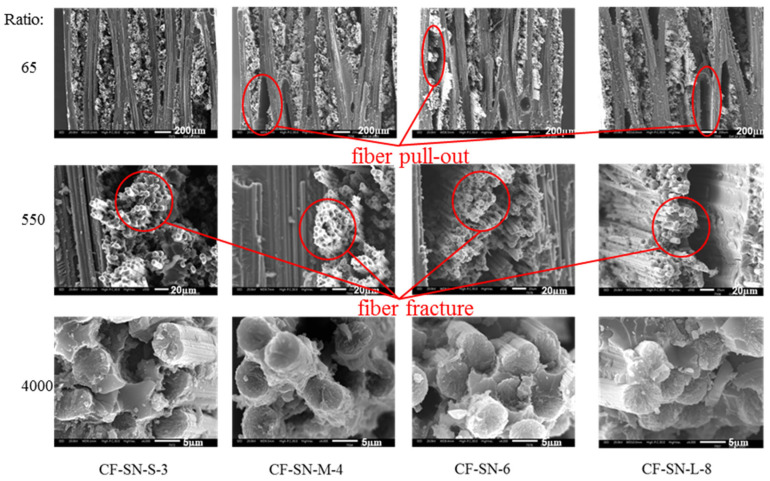
SEM images of the fracture position of the specimen group with an equal ratio of W/2C; no significant difference in the failure modes for the specimens of this group. Failure modes present mostly in fiber fracture and a small part of fiber pull-out and delamination, with a relatively smooth fiber fracture face.

**Figure 15 polymers-14-05552-f015:**
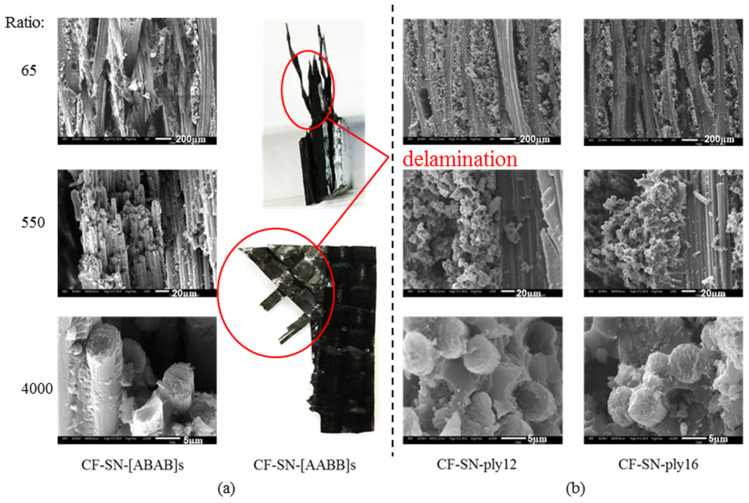
SEM images of the fracture position of the specimen group (**a**) with different lay-up sequences and (**b**) with different layer numbers. The specimen containing 45° fiber orientation was pulled out and the lay-up was delaminated severely, and small pieces of broken fiber can be seen in thicker specimens.

**Figure 16 polymers-14-05552-f016:**
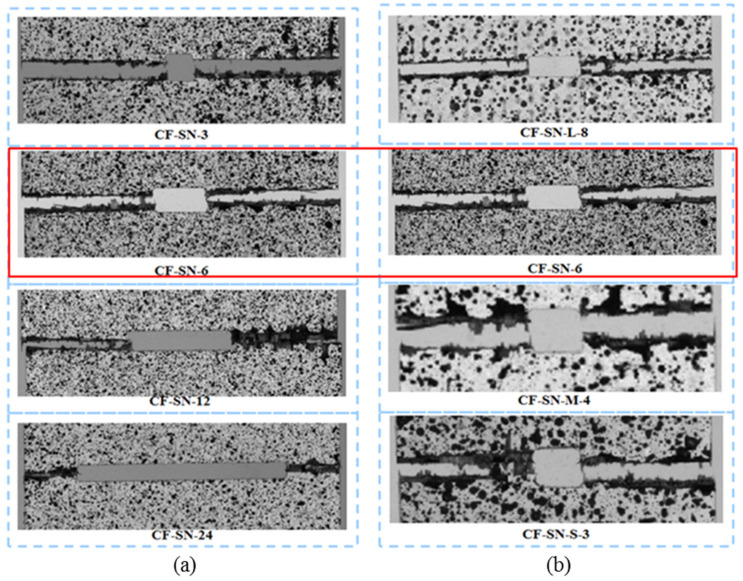
Local failure characteristic diagram of tensile specimens (**a**) with equal specimen size and (**b**) with an equal ratio of W/2C. No significant difference in the failure modes for specimens of this two groups; failure modes present mostly in fiber fracture and a small part of fiber pull-out and delamination.

**Figure 17 polymers-14-05552-f017:**
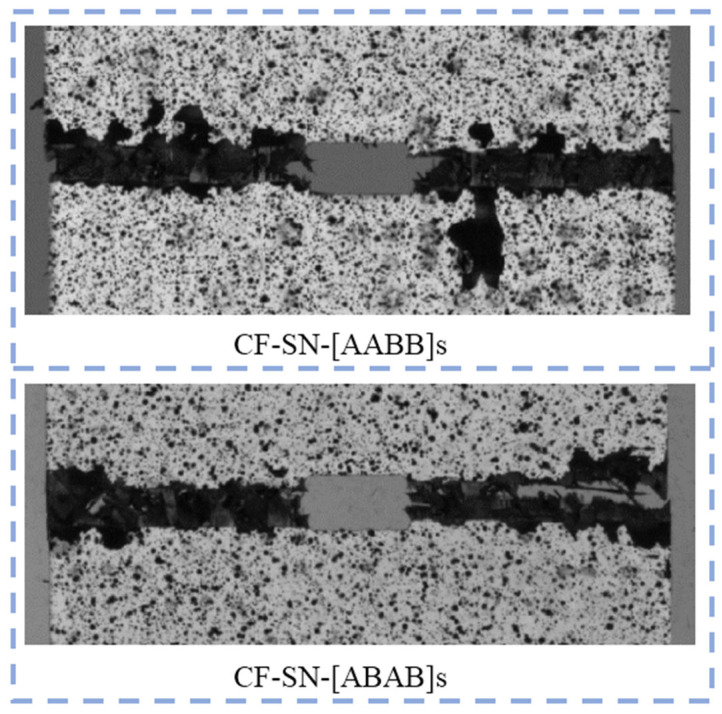
Local failure characteristic diagram of tensile specimens with different lay-up sequences; the outer plain fabric of the CF-SN-[AABB]s specimen was found to be cracked first, and the internal ±45° layer failed before the whole specimen was completely broken. In contrast, the CF-SN-[ABAB]s specimen fractured almost completely with more evenly distributed stress.

**Figure 18 polymers-14-05552-f018:**
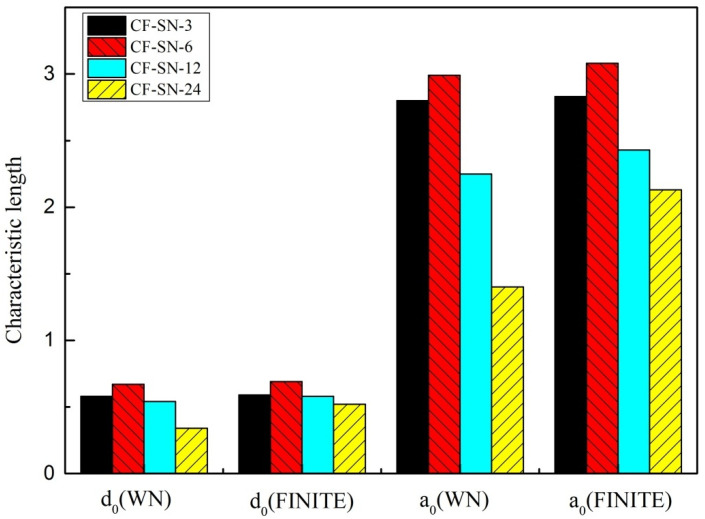
Characteristic lengths of the equal specimen size group calculated by the WN stress criteria and finite-width stress model; the characteristic lengths in this group calculated under the WN-stress-criteria-based formula vary greatly, especially with a greater notch size.

**Figure 19 polymers-14-05552-f019:**
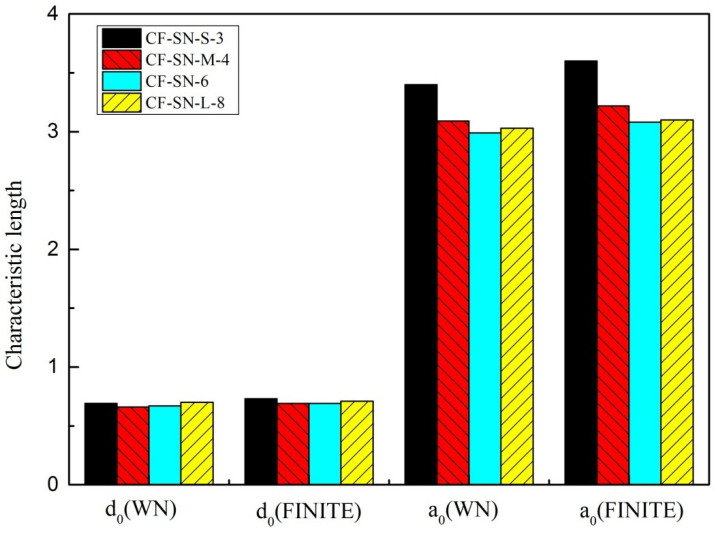
Characteristic lengths of the equal ratio (W/2C) group calculated by the WN stress criteria and finite-width stress model; the size ratio of the notch to the specimen remained at 1:6 (which can be approximately regarded as an infinite-width plate), so the characteristic length of each specimen in this group is fairly consistent.

**Figure 20 polymers-14-05552-f020:**
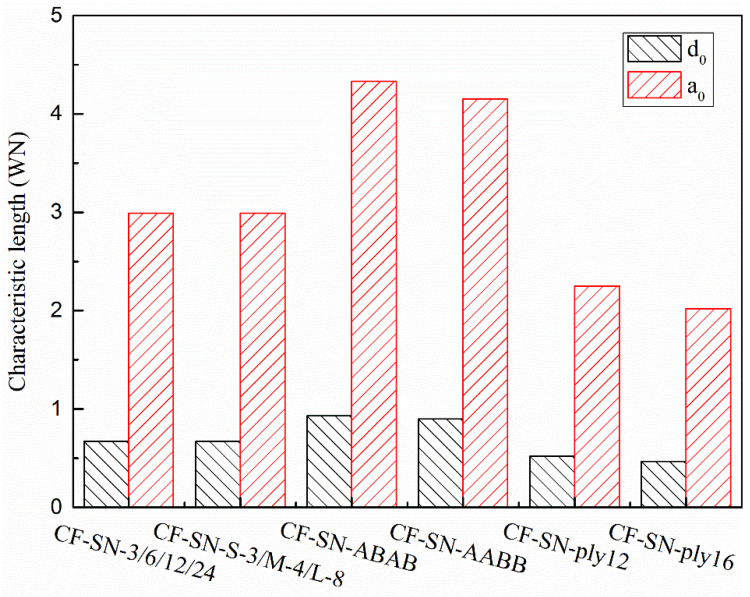
Average characteristic lengths, $d_{0}$ and $a_{0}$, of each type of specimen (equal specimen size group/equal ratio of W/2C group/CF-SN-ABAB/CF-SN-AABB/CF-SN-ply12/CF-SN-ply16) calculated by the WN stress criteria; both the lay-up sequence and layer number affect the characteristic length.

**Figure 21 polymers-14-05552-f021:**
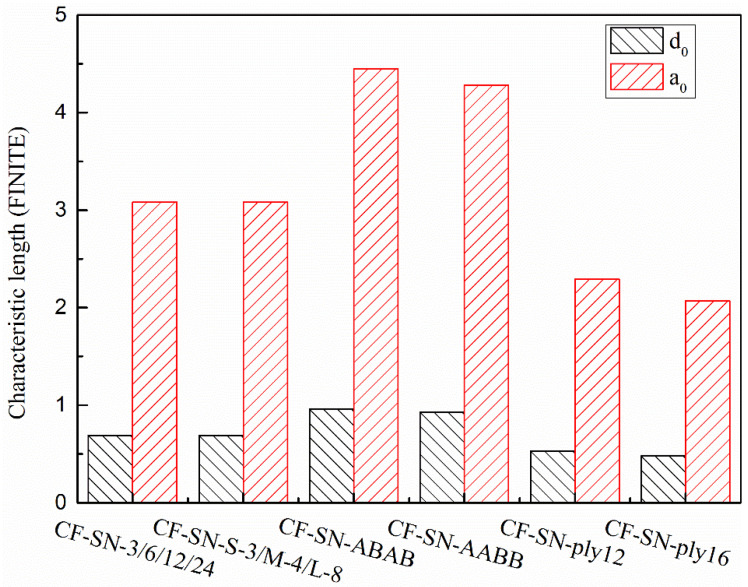
Average characteristic lengths, $d_{0}$ and $a_{0}$, of each type of specimen (equal specimen size group/equal ratio of W/2C group/CF-SN-ABAB/CF-SN-AABB/CF-SN-ply12/CF-SN-ply16) calculated by the finite-width stress model; both the lay-up sequence and layer number affect the characteristic length.

**Figure 22 polymers-14-05552-f022:**
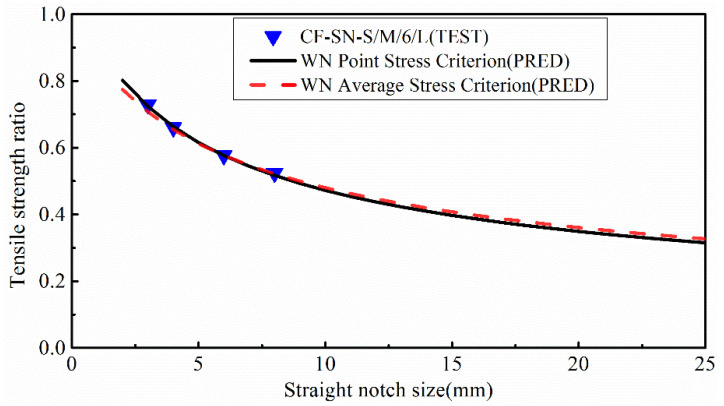
Comparison of the tensile strength ratio curves of the equal W/2C ratio group predicted by the point stress criterion and average stress criterion in the WN stress criteria with test results; it meets the curve of the test results well.

**Figure 23 polymers-14-05552-f023:**
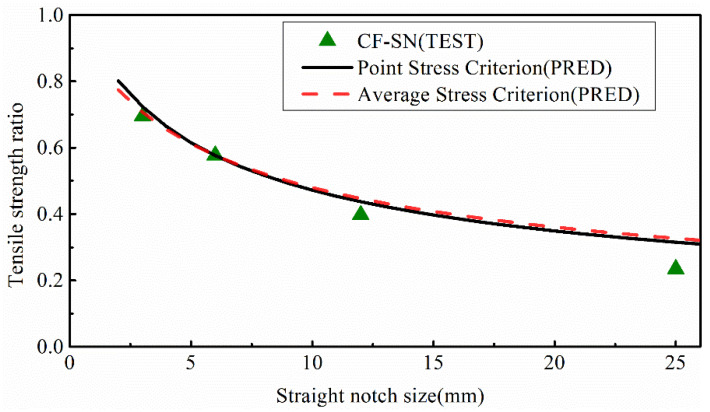
Comparison of the tensile strength ratio curves of the equal specimen size group predicted by the point stress criterion and average stress criterion in the WN stress criteria with test results; the predicted results are not in good agreement with the experimental results when W/2C<6.

**Figure 24 polymers-14-05552-f024:**
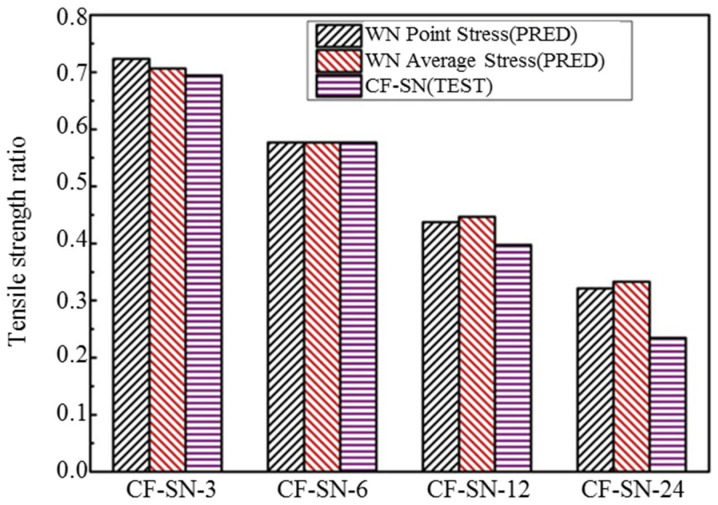
Comparison of the tensile strength of the equal specimen size group predicted by the point stress criterion and average stress criterion in the WN stress criteria with test results. The predicted results are not in good agreement with the experimental results when W/2C<6.

**Figure 25 polymers-14-05552-f025:**
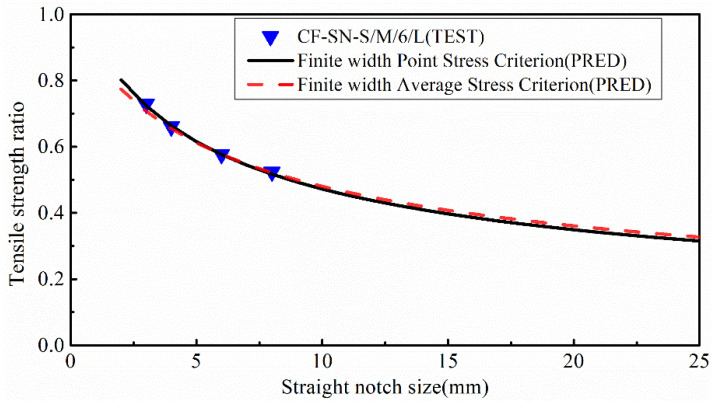
Comparison of the tensile strength ratio curves of the equal W/2C ratio group predicted by the point stress criterion and average stress criterion in the finite-width stress criterion with test results; it meets the curve of the test results well.

**Figure 26 polymers-14-05552-f026:**
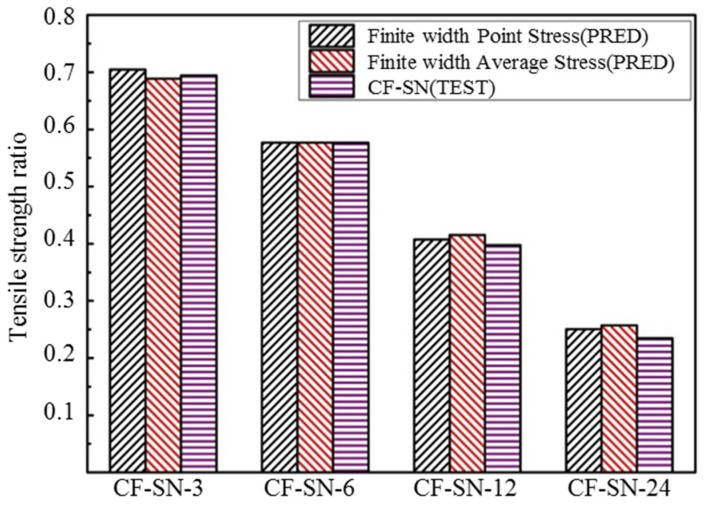
Comparison of the tensile strength of the equal specimen size group predicted by the point stress criterion and average stress criterion in the finite-width stress criterion with test results; compared with the WN stress criterion, the predicted value of the new formula is closer to the real experimental values, especially when the notch size is large (W/2C≤6).

**Figure 27 polymers-14-05552-f027:**
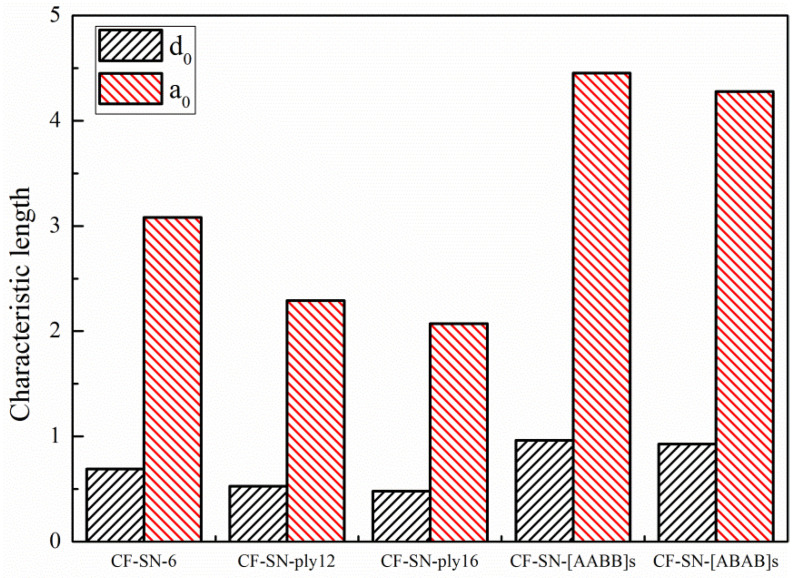
Comparison of characteristic lengths calculated by the standard specimen of each type of specimen (equal specimen size group/equal ratio of W/2C group/CF-SN-ABAB/CF-SN-AABB/CF-SN-ply12/CF-SN-ply16); the characteristic length is different when the class of specimen (lay-up sequence, and same lay-up thickness) is different.

**Figure 28 polymers-14-05552-f028:**
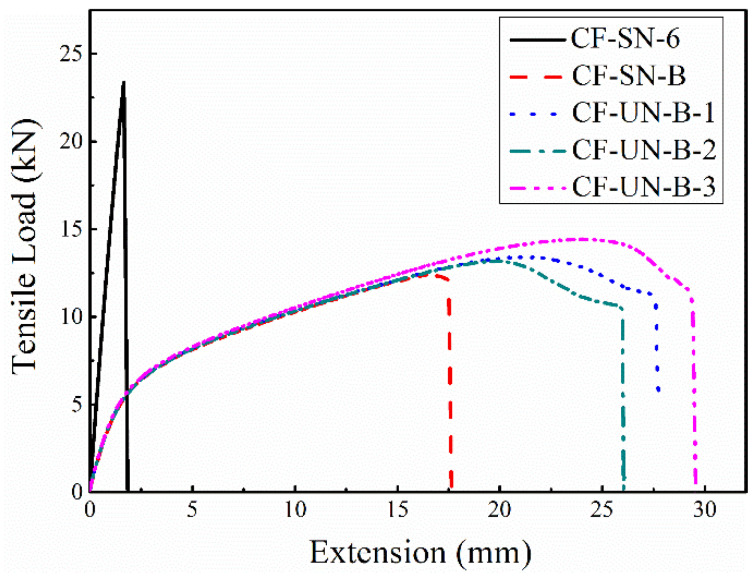
Comparison of the tensile load–displacement curves between specimens with a ±45° layer (CF-SN-B) and specimens with a 0°/90° layer (CF-SN-6); the load–displacement curves of different non-notched specimens are quite different when the specimens were loaded to more than 95% of the peak load.

**Table 1 polymers-14-05552-t001:** Finite-width specimens for experimental tests.

No.	Size L×W (mm)	Thickness (mm)	Crack Length 2C (mm)	Ply Angle and Sequence
CF-SN-3	250×36	2	3	[0°/90°]_8_
CF-SN-6	250×36	2	6	[0°/90°]_8_
CF-SN-12	250×36	2	12	[0°/90°]_8_
CF-SN-24	250×36	2	24	[0°/90°]_8_
CF-SN-S-3	125×18	2	3	[0°/90°]_8_
CF-SN-M-4	150×24	2	4	[0°/90°]_8_
CF-SN-L-8	300×48	2	8	[0°/90°]_8_
CF-SN-[ABAB]s	250×36	2	6	[(0°/90°)/±45°]_2S_
CF-SN-[AABB]s	250×36	2	6	[(0°/90°)/(0°/90°)/±45°/±45°]_S_
CF-SN-ply12	250×36	3	6	[0°/90°]_12_
CF-SN-ply16	250×36	4	6	[0°/90°]_16_

**Table 2 polymers-14-05552-t002:** Tensile strength ratios and notch sensitivity.

Specimens	Peak Load Ratio	Notch Sensitivity	Specimens	Peak Load Ratio	Notch Sensitivity
CF-SN-3	0.69457	1.43974	CF-SN-S-3	0.72896	1.37182
CF-SN-6	0.57672	1.73394	CF-SN-M-4	0.66023	1.51462
CF-SN-12	0.39746	2.51598	CF-SN-6	0.57672	1.73394
CF-SN-24	0.23459	4.26276	CF-SN-L-8	0.52393	1.90865
CF-SN-[AABB]s	0.64553	1.54911	CF-SN-ply12	0.52111	1.91898
CF-SN-[ABAB]s	0.63808	1.56720	CF-SN-ply16	0.50224	1.99108

**Table 3 polymers-14-05552-t003:** The coefficient of variation of characteristic length.

	Name	$\mathrm{SD}\;\left(d_{0}\right)$	$\mathrm{SD}\;\left(a_{0}\right)$	$\mathrm{cv}\%\;\left(d_{0}\right)$	$\mathrm{cv}\%\;\left(a_{0}\right)$
WN					
Equation size		0.12	0.62	0.22	0.26
Equation ratio		0.01	0.16	0.02	0.05
**Finite**					
Equation size		0.06	0.36	0.10	0.14
Equation ratio		0.02	0.21	0.02	0.06

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
