# Peer review of "A New Stress-Based Formulation for Modeling Notched Fiber-Reinforced Laminates"

_polymers, 2022, doi:10.3390/polym14245552_

Round 1

Reviewer 1 Report

This paper experimentally and numerically investigated the stress concentration of notch edge. The quality of figures is very good. The paper can be suggested for publication after the revision.

1- Please add a notation list before Introduction Section. For examples “???”, “?”,CF-SN-[AABB]s”, ..

2- In order to provide a more comprehensive literature review, the authors should cite and discuss the following relevant paper in their revised manuscript:

Stress concentration factors in tubular X-connections retrofitted with FRP under compressive load. Ocean Engineering 229 (2021): 108562.

3- In section 4.5 , we see “Fig. 13-Fig. 15.” Please revise to “Figs. 13-15.” Please revise similar issues.

4- The thickness of each layer is 0.25 mm. Why is this value selected?

Author Response

Response to Reviewer 1 Comments

Point 1: Please add a notation list before Introduction Section. For examples “???”, “?”, “CF-SN-[AABB]s”,

Response 1: The notation list has been added in the article.

Point 2: In order to provide a more comprehensive literature review, the authors should cite and discuss the following relevant paper in their revised manuscript:

Stress concentration factors in tubular X-connections retrofitted with FRP under compressive load. Ocean Engineering 229 (2021): 108562.

Response 2: The relevant paper has been cited and discussed in the article and highlighted.

Point 3: In section 4.5 , we see “Fig. 13-Fig. 15.” Please revise to “Figs. 13-15.” Please revise similar issues.

Response 3: All similar issues have been modified accordingly. It has been marked in the article.

Point 4: The thickness of each layer is 0.25 mm. Why is this value selected?

Response 4: This thickness is slightly greater than the superposition thickness of the two bundles after plain weaving and can more easily constitute the required integer thickness of the majority situation. This is also the most common thickness of the carbon fiber prepreg from the company we purchased them. The actual thickness of each layer will be slightly different according to the processing situation, but the overall thickness can be controlled.

Reviewer 2 Report

There are some weaknesses through the manuscript which need improvement. Therefore, the submitted manuscript cannot be accepted for publication in this form, but it has a chance of acceptance after a major revision. My comments and suggestions are as follows:

1- Abstract gives information on the main feature of the performed study, but some details about the obtained results must be added. However, a concise abstract is needed.

2- First part (a couple of sentences) of abstract gives general information which should be removed.

3- Authors must clarify necessity of the performed research. Aims and objectives of the study, and also differences with the previous review papers must be clearly mentioned.

4- The literature study must be enriched. For instance, authors must read and refer to the relevant papers: (a) https://doi.org/10.1016/j.jmrt.2022.05.068 (b) https://doi.org/10.1016/j.engfracmech.2022.108724 and other research works.

5- Authors must clearly emphasized the strengths and limitations of their study/methods.

6- The main reference of formula must be cited.

7- How the crack lengths are selected? Why these particular crack lengths are considered in this study?

8- It would be great, if authors provide a real figure of specimens before and after applied load.

9- It is necessary to add stress-strain relationship of the examined specimens (presenting only force-displacement curves is not enough).

10- The conclusion must be more than just a summary of the manuscript. List of references must be updated based on the proposed papers. Please provide all changes by red color in the revised version.

Author Response

Response to Reviewer 2 Comments

Point 1: Abstract gives information on the main feature of the performed study, but some details about the obtained results must be added. However, a concise abstract is needed.

Response 1: The most significant result of this study was already pointed out at the end of the abstract. some details about the obtained results has be added.

Point 2: First part (a couple of sentences) of abstract gives general information which should be removed.

Response 2: The abstract has been adjusted in the article.

Point 3: Authors must clarify necessity of the performed research. Aims and objectives of the study, and also differences with the previous review papers must be clearly mentioned.

Response 3: The necessity, research objectives and uniqueness with the previous work have been further clarified in the introduction section. Among them, the main difference is that the previous research on open-hole fiber reinforced materials mainly focuses on the study of infinite width plate, this has been explained in the last paragraph of the introduction.

Point 4: The literature study must be enriched. For instance, authors must read and refer to the relevant papers:

(a) https://doi.org/10.1016/j.jmrt.2022.05.068

(b) https://doi.org/10.1016/j.engfracmech.2022.108724

and other research work

Response 4: More references, including those mentioned above, have been analyzed and cited in the literature study.

Point 5: Authors must clearly emphasized the strengths and limitations of their study/methods.

Response 5: The strengths and limitations have been further clarified in the Article.

Point 6: The main reference of formula must be cited.

Response 6: The sources of all formulas have been cited in their descriptive paragraphs, because a series of formulas for calculating a parameter often have the same source, so they are not quoted separately behind each formula, but quoted as a whole in the descriptive paragraph.

Point 7: How the crack lengths are selected? Why these particular crack lengths are considered in this study?

Response 7: The crack size of standard specimen is selected according to the critical value of the theoretical basis mentioned in section 2 that when W/2C≥6, the test specimen can be regarded as infinite width. According to ASTM test standard, the width of the standard specimen is 36mm (W), so the crack size of the standard specimen is equal to W/6=6mm (same in ASTM D5766). The overall size of other specimens and the selection of crack sizes are determined according to the needs of comparative experiments. A more specific explanation of crack size selection has been added in 3.1.

Point 8: It would be great, if authors provide a real figure of specimens before and after applied load.

Response 8: We are sorry that due to confidentiality, only processed experimental data can be provided in this study. However, the carbon fiber prepreg (T300, 3K) used in this study was supplied by Wei hai Guangwei Group Co., Ltd (Related instructions have been added to Section 3). In addition, the specific size and processing mode of the specimen as well as the specific experimental mode and machine model have been given. If you are interested, you can repeat the experiment or extend the experiment and calculate the theoretical value by using the formula in this article for reliability verification.

Point 9: It is necessary to add stress-strain relationship of the examined specimens (presenting only force-displacement curves is not enough).

Response 9: The cross-sectional area of the specimen in this study does not change significantly before failure, so the stress-strain curve can be directly obtained from the force-displacement curve, and there is no significant difference between the analysis results and the slope of the force-displacement curve. There is no need to repeat the analysis. In addition, the theoretical model proposed in this study is mainly used to predict the ultimate strength of fiber-reinforced materials.

Point 10: The conclusion must be more than just a summary of the manuscript. List of references must be updated based on the proposed papers. Please provide all changes by red color in the revised version.

Response 10: The conclusion has been modified and adjusted accordingly, new references have been added to the references list , and all the above problems have been modified by red color in the revised version.

Round 2

Reviewer 1 Report

The paper is ready for publication.

Reviewer 2 Report

The paper has been improved and corresponding modifications have been conducted. In my opinion, the current version can be considered for publication.